# On Prior Distributions and Approximate Inference for Structured Variables

**Oluwasanmi Koyejo**
Psychology Dept., Stanford
sanmi@stanford.edu

**Rajiv Khanna**
ECE Dept., UT Austin
rajivak@utexas.edu

**Joydeep Ghosh**
ECE Dept., UT Austin
ghosh@ece.utexas.edu

**Russell A. Poldrack**
Psychology Dept., Stanford
poldrack@stanford.edu

## Abstract

We present a general framework for constructing prior distributions with structured variables. The prior is defined as the information projection of a base distribution onto distributions supported on the constraint set of interest. In cases where this projection is intractable, we propose a family of parameterized approximations indexed by subsets of the domain. We further analyze the special case of sparse structure. While the optimal prior is intractable in general, we show that approximate inference using convex subsets is tractable, and is equivalent to maximizing a submodular function subject to cardinality constraints. As a result, inference using greedy forward selection provably achieves within a factor of (1-1/e) of the optimal objective value. Our work is motivated by the predictive modeling of high-dimensional functional neuroimaging data. For this task, we employ the Gaussian base distribution induced by local partial correlations and consider the design of priors to capture the domain knowledge of sparse support. Experimental results on simulated data and high dimensional neuroimaging data show the effectiveness of our approach in terms of support recovery and predictive accuracy.

## 1 Introduction

Data in scientific and commercial disciplines are increasingly characterized by high dimensions and relatively few samples. For such cases, a-priori knowledge gleaned from expertise and experimental evidence are invaluable for recovering meaningful models. In particular, knowledge of restricted degrees of freedom such as sparsity or low rank has become an important design paradigm, enabling the recovery of parsimonious and interpretable results, and improving storage and prediction efficiency for high dimensional problems. In Bayesian models, such restricted degrees of freedom can be captured by incorporating structural constraints on the design of the prior distribution. Prior distributions for structured variables can be designed by combining conditional distributions - each capturing portions of the problem structure, into a hierarchical model. In other cases, researchers design special purpose prior distributions to match the application at hand. In the case of sparsity, an example of the former approach is the *spike and slab* prior [1, 2], and an example of the latter approach is the *horseshoe* prior [3].

We describe a framework for designing prior distributions when the a-priori information include structural constraints. Our framework follows the *maximum entropy principle* [4, 5]. The distribution is chosen as one that incorporates known information, but is as difficult as possible to discriminate from the base distribution with respect to relative entropy. The maximum entropy approach

has been especially successful with domain knowledge expressed as expectation constraints. In such cases, the solution is given by a member of the exponential family [6, 7] e.g. quadratic constraints result in the Gaussian distribution. Our work extends this framework to the design of prior distributions when a-priori information include domain constraints.

Our main technical contributions are as follows:

- We show that under standard assumptions, the information projection of a base density to domain constraints is given by its restriction (Section 2).
- We show the equivalence between relative entropy inference with data observation constraints and Bayes rule for continuous variables
- When such restriction is intractable, we propose a family of parameterized approximations indexed by subsets of the domain (Section 2.1).

We consider approximate inference in the special case of sparse structure:

- We characterize the restriction precisely, showing that it is given by a conditional distribution (Section 3).
- We show that the approximate sparse support estimation problem is submodular. As a result, greedy forward selection is efficient and guarantees $(1-\frac{1}{e})$ factor optimality (Section 3.1).

Our work is motivated by the predictive modeling of high-dimensional functional neuroimaging data, measured by cognitive neuroscientists for analyzing the human brain. The data are represented using hundreds of thousands of variables. Yet due to real world constraints, most experimental datasets contain only a few data samples [8]. The proposed approach is applied to predictive modeling of simulated data and high-dimensional neuroimaging data, and is compared to Bayesian hierarchical models and non-probabilistic sparse predictive models, showing superior support recovery and predictive accuracy (Section 4). Due to space constraints, all proofs are provided in the supplement.

## 1.1 Preliminaries

This section includes notation and a few basic definitions. Vectors are denoted by lower case $x$ and matrices by capital $\mathbf{X}$. $x_{i,j}$ denotes the $(i,j)^{th}$ entry of the matrix $\mathbf{X}$. $x_{i,:}$ denotes the $i^{th}$ row of $\mathbf{X}$ and $x_{:,j}$ denotes the $j^{th}$ column. Let $|\mathbf{X}|$ denote the determinant of $\mathbf{X}$. Sets are denoted by sans serif e.g. S. The reals are denoted by $\mathfrak{R}$. $[n]$ denotes the set of integers $\{1, \ldots, n\}$, and $\wp(n)$ denotes the power set of $[n]$. Let X be either a countable set, or a complete separable metric space equipped with the standard Borel $\sigma$-algebra of measurable set. Let $\mathcal{P}$ denote the set of probability densities on X i.e. positive functions $\mathcal{P} = \{p : \mathsf{X} \mapsto [0,1] , \int_{\mathsf{X}} p(x) = 1\}$. For the remainder of this paper, we make the following assumption:

**Assumption 1.** *All distributions $\boldsymbol{P}$ are absolutely continuous with respect to the dominating measure $\nu$ so there exists a density $p \in \mathcal{P}$ that satisfies $d\boldsymbol{P} = pd\nu$.*

To simplify notation, we use use the standard $d\nu = dx$. As a consequence of Assumption 1, the *relative entropy* is given in terms of the densities as:

$$\mathrm{KL}(q\|p) = \int_{\mathsf{X}} q(x) \log \frac{q(x)}{p(x)} dx.$$

The relative entropy is strictly convex with respect to its first argument. The *information projection* of a probability density $p$ to a constraint set $\mathcal{A}$ is given by the solution of:

$$\inf_{q \in P} \mathrm{KL}(q\|p) \text{ s.t. } q \in \mathcal{A}.$$

We will only consider projections where $\mathcal{A}$ is a closed convex set so the infimum is achieved. The *delta function*, denoted by $\delta_{(\cdot)}$, is a generalized set function that satisfies $\int_{\mathsf{X}} \delta_{\mathsf{A}}(x) f(x) dx = \int_{\mathsf{A}} f(x) dx$, and $\int_{\mathsf{X}} \delta_{\mathsf{A}}(x) dx = 1$, for some some $\mathsf{A} \subseteq \mathsf{X}$. The set of *domain restricted densities*, denoted by $\check{\mathcal{F}}_{\mathsf{A}}$ for $\mathsf{A} \subset \mathsf{X}$, is the set of probability density functions supported on A i.e.

$\mathcal{F}_\mathsf{A} = \{q \in \mathcal{P} \mid q(x) = 0 \; \forall \; x \notin \mathsf{A}\} \cup \{\delta_{\{x\}} \; \forall \; x \in A\} \subset \mathcal{F}_\mathsf{A} \subset \mathcal{P} = \mathcal{F}_\mathsf{X}$. Further, note that $\mathcal{F}_\mathsf{A}$ is closed and convex for any $\mathsf{A} \subseteq \mathsf{X}$ (including nonconvex A).

*Restriction* is a standard approach for defining distributions on subsets $\mathsf{A} \subseteq \mathsf{X}$. An important special case we will consider is when A is a measure zero subset of X. The common conditional density is one such example, the existence of which follows from the *disintegration theorem* [9]. Restrictions of measure require extensive technical tools in the general case [10]. We will employ the following simplifying condition for the remainder of this manuscript:

**Condition 2.** *The sample space* $\mathsf{X}$ *is a subset of Euclidean space with $\nu$ given by the Lebesgue measure. Alternatively,* $\mathsf{X}$ *is a countable set with $\nu$ given by the counting measure.*

Let $\boldsymbol{P}$ be a probability distribution on $\mathsf{X}$. Under Assumption 1 and Condition 2, the restriction of the density $p$ to the set $\mathsf{A} \subset \mathsf{X}$ is given by:

$$q(x) = \begin{cases} \frac{p(x)}{\int_\mathsf{A} p(x) dx} & x \in \mathsf{A}, \\ 0 & \text{otherwise.} \end{cases}$$

## 2 Priors for structured variables

We assume a-priori information identifying the structure of $X$ via the sub-domain $\mathsf{A} \subset \mathsf{X}$. We also assume a pre-defined base distribution $\boldsymbol{P}$ with associated density $p$. Without loss of generality, let $p$ have support everywhere[1] on $\mathsf{X}$ i.e. $p(x) > 0 \; \forall \; x \in \mathsf{X}$. Following the *principle of minimum discrimination information*, we select the prior as the *information projection* of the base density $p$ to $\mathcal{F}_\mathsf{A}$. Our first result identifies the equivalence between information projection subject to domain constraints and density restriction.

**Theorem 3.** *Under Condition 2, the information projection of the density $p$ to the constraint set $\mathcal{F}_\mathsf{A}$ is the restriction of $p$ to the domain* A.

Theorem 3 gives principled justification for the domain restriction approach to structured prior design. Examples of density restriction in the literature include the truncated Gaussian, Beta and Gamma densities [11], and the restriction of the matrix-variate Gaussian to the manifold of low rank matrices [12]. Various properties of the restriction, such as its shape, and tail behavior (up to re-scaling) follow directly from the base density. Thus the properties of the resulting prior are more amenable to analysis when the base measure is well understood. Next, we consider a corollary of Theorem 3 that was introduced by Williams [13].

**Corollary 4.** *Consider the product space* $\mathsf{X} = \mathsf{W} \times \mathsf{Y}$. *Let domain constraint be given by* $\mathsf{W} \times \{\hat{y}\}$ *for some* $\hat{y} \in \mathsf{Y}$. *Under Condition 2, the information projection of $p$ to $\mathcal{F}_{\mathsf{W} \times \{\hat{y}\}}$ is given by* $p(w|\hat{y})\delta_{\hat{y}}$.

In the Bayesian literature, $p(w)$ is known as the prior, $p(y|w)$ is the likelihood and $p(w|\hat{y})$ is the posterior density given the observation $y = \hat{y}$. Corollary 4 considers the information projection of the joint density $p(w, y)$ given observed data, and shows that the solution recovers the Bayesian posterior. Williams [13] considered a generalization of Corollary 4, but did not consider projection to data constraints[2]. While Corollary 4 has been widely applied in the literature e.g. [14], to the best of our knowledge, the presented result is the first formal proof.

### 2.1 Approximate inference for structured variables via tractable subsets

For many structural constraints of interest, restriction requires the computation of an intractable normalization constant. In theory, rejection sampling and Markov Chain Monte Carlo (MCMC) inference methods [15] do not require normalized probabilities. However, as many structured sub-domains are measure zero sets with respect to the dominating measure, randomly generated samples generated from the base distribution are unlikely to lie in the constrained domains e.g. random samples from a multivariate Gaussian are not sparse. Hence rejection sampling fails, and MCMC suffers from low acceptance probabilities. As a result, inference on such structured sub-domains

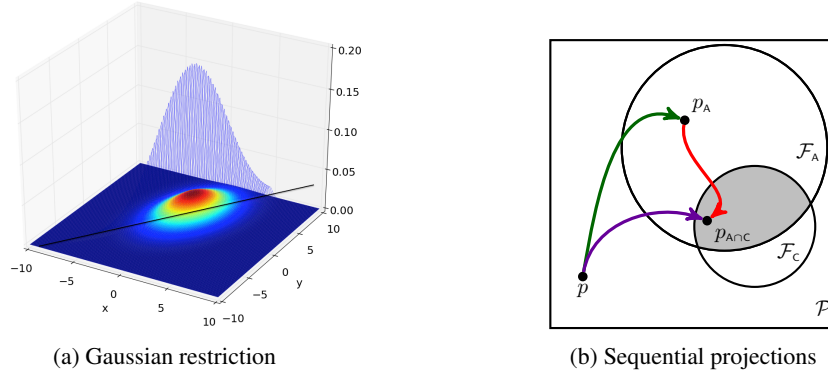

| (a) Gaussian restriction | (b) Sequential projections |

Figure 1: **(a)** Gaussian density and restriction to diagonal line shown. **(b)** Illustration of Theorem 5; sequence of information projections $\mathcal{P} \to \mathcal{F}_A \to \mathcal{F}_C$ and $\mathcal{P} \to \mathcal{F}_{A \cap C}$ are equivalent.

typically requires specialized methods e.g. [11, 12]. In the following, we propose a class of variational approximations based on an inner representation of the structured subdomain. Let $\{S_i \in A\}$ represent a (possibly overlapping) partitioning of $A$ into subsets. We define the domain restricted density sets generated by these partitions as $\mathcal{F}_{S_i}$, and their union $\mathcal{D} = \bigcup \mathcal{F}_{S_i}$. Note that by definition each $\mathcal{F}_{S_i} \subseteq \mathcal{D} \subseteq \mathcal{F}_A \subseteq \mathcal{F}_X$. Our approach is to approximate the optimization over densities in $\mathcal{F}_A$ by optimizing over $\mathcal{D}$ - a smaller subset of tractable densities.

Approximate inference is generally most successful when the approximation accounts for observed data. Inspired by the results of Corollary 4, we consider such a projection. Let $p_A(w, y)$ be the information projection of the joint distribution $p(x, y)$ to the set $\mathcal{F}_{A \times \{\hat{y}\}}$. We propose approximate inference via the following rule:

$$p_{S_*, \hat{y}} = \underset{q \in \mathcal{D} \times \mathcal{F}_{\{\hat{y}\}}}{\arg \min} \ \mathrm{KL}(q(w, y) \| p_A(w, y)) = \underset{S}{\arg \min} \left[ \underset{q \in \mathcal{F}_{S \times \{\hat{y}\}}}{\min} \ \mathrm{KL}(q(w, y) \| p_A(w, y)) \right]. \quad (1)$$

Our proposed approach may be decomposed into two steps. The inner step is solved by estimating a parameterized set of prior densities $\{q_S\}$ corresponding to choices of $S$, and the outer step is solved by the selection of the optimal subset $S_*$. The solution is given by $p_{S_*, \hat{y}}(w, y) = p_{S_*}(w|\hat{y}) \delta_{\hat{y}}$ (Corollary 4) with the associated approximate posterior given by $p_{S_*}(w|\hat{y})$.

The following theorem considers the effect of a sequence of domain constrained information projections (see Fig. 1b), which will useful for subsequent results.

**Theorem 5.** *Let $\pi : [n] \mapsto [n]$ be a permutation function and $\{C_{\pi(i)} \mid C_{\pi(i)} \subset X\}$ represent a sequence of sets with non empty intersection $B = \bigcap C_i \neq \emptyset$. Given a base density $p$, let $q_0 = p$, and define the sequence of information projections:*

$$q_i = \underset{q \in \mathcal{F}_{C_{\pi(i)}}}{\arg \min} \ \mathrm{KL}(q \| q_{i-1}).$$

*Under Condition 2, $q_* = q_N$ is independent of $\pi$. Further $q_* = \underset{q \in \mathcal{F}_B}{\min} \ \mathrm{KL}(q \| p)$.*

We apply Theorem 5 to formulate equivalent solutions of (1) that may be simpler to solve.

**Corollary 6.** *Let $p_{S_*, \hat{y}}(w, y)$ be the solution of (1), then the posterior distribution $p_{S_*}(w|\hat{y})$ is given by:*

$$p_{S_*}(w|\hat{y}) = \underset{q \in \mathcal{D}}{\arg \min} \ \mathrm{KL}(q(w) \| p_A(w|\hat{y})) = \underset{q \in \mathcal{D}}{\arg \min} \ \mathrm{KL}(q(w) \| p(w|\hat{y})). \quad (2)$$

Corollary 6 implies that we can estimate the approximate structured posterior directly as the information projection of the unstructured posterior distribution $p(w|\hat{y})$. Upon further examination, Corollary 6 also suggests that the proposed approximation is most useful when there exist subsets of $A$ such that the restriction of the base density to each subset leads to tractable inference. Further, the result is most accurate when one of the subsets $S_* \in A$ captures most of the posterior probability mass. When the optimal subset $S_*$ is known, the structured prior density associated with the structured posterior can be computed as shown in the following corollary.

**Corollary 7.** *Let $p_{\mathsf{s}_*,\hat{y}}(w, y)$ be the solution of* (1). *Define the density $p_{\mathsf{s}_*}(w)$ as:*

$$p_{\mathsf{s}_*}(w) = \arg\min_{q \in \mathcal{F}_{\mathsf{s}_*}} \mathrm{KL}(q(w) \| p_{\mathsf{A}}(w)) = \arg\min_{q \in \mathcal{F}_{\mathsf{s}_*}} \mathrm{KL}(q(w) \| p(w)). \tag{3}$$

*then $p_{\mathsf{s}_*}(w)$ is the prior distribution corresponding to the Bayesian posterior $p_{\mathsf{s}_*}(w|\hat{y})$.*

## 3 Priors for sparse structure

We now consider a special case of the proposed framework for sparse structured variables. A $d$ dimensional variable $\boldsymbol{x} \in \mathsf{X}$ is *k-sparse* if $d - k$ of its entries take a *default* value of $c_i$ i.e $|\{i \mid x_i = c_i\}| = d - k$. In Euclidean space $\mathsf{X} = \mathfrak{R}^d$ and in most cases, $c_i = 0 \ \forall \ i$. Similarly, the distribution $\boldsymbol{P}$ on the domain $\mathsf{X}$ is $k$-sparse if all random variables $X \sim \boldsymbol{P}$ are at most $k$-sparse. The *support* of $\boldsymbol{x} \in \mathsf{X}$ is the set $supp(\boldsymbol{x}) = \{i \mid x_i \neq c_i\} \in \wp(d)$. Let $\mathsf{S} \subset \mathsf{X}$ denote the set of variables with support $\mathsf{s}$ i.e. $\mathsf{S} = \{\boldsymbol{x} \in \mathsf{X} \text{ s.t. } supp(\boldsymbol{x}) = \mathsf{s}\}$. We will use the notation $\boldsymbol{x}_{\mathsf{s}} = \{x_i \mid i \in \mathsf{s}\}$, and its complement $\boldsymbol{x}_{\mathsf{s}'} = \{x_i \mid i \in \mathsf{s}'\}$, where $\mathsf{s}' = [d]\backslash\mathsf{s}$. The domain of $k$ sparse vectors is given by the union of all possible $\frac{d!}{(d-k)!k!}$ sparse support sets as $\mathsf{A} = \bigcup \mathsf{S}_i$. While the sparse domain $\mathsf{A}$ is non-convex, each subset $\mathsf{S}$ is a convex set, in fact given by linear subspaces with basis $\{e_i \mid i \in \mathsf{s}\}$. Further, while the information projection of a base density $p$ to $\mathsf{A}$ is generally intractable, the information projection to its convex subsets $\mathsf{S}$ turn out to be computationally tractable. We investigate the application of the proposed approximation scheme using these subsets.

Consider the information projection of an arbitrary probability measure $\boldsymbol{P}$ with density[3] $p$ to the set $\mathcal{D} = \bigcup \mathcal{F}_{\mathsf{s}_i}$ given by:

$$\min_{q \in \mathcal{D}} \mathrm{KL}(q \| p) = \min_{\mathsf{S} \in \{\mathsf{S}_i\}} \left[ \min_{q \in \mathcal{F}_{\mathsf{S}}} \mathrm{KL}(q \| p) \right] = \min_{\mathsf{S} \in \{\mathsf{S}_i\}} \mathrm{KL}(p_{\mathsf{s}} \| p).$$

Applying Theorem 3, we can compute that $p_{\mathsf{s}} = p(\boldsymbol{x})\delta_{\mathsf{s}}(\boldsymbol{x})/Z$, where $Z$ is a normalization factor:

$$Z = \int_{\mathsf{S}} p(\boldsymbol{x}) = \int_{\mathsf{X}} p(\boldsymbol{x}_{\mathsf{s}}, \boldsymbol{x}_{\mathsf{s}'})\delta_{\mathsf{s}}(\boldsymbol{x}) = \int_{\mathsf{X}} p(\boldsymbol{x}_{\mathsf{s}}|\boldsymbol{x}_{\mathsf{s}'})p(\boldsymbol{x}_{\mathsf{s}'})\delta_{\mathsf{s}}(\boldsymbol{x}) = p(\boldsymbol{x}_{\mathsf{s}'} = \boldsymbol{c}_{\mathsf{s}'}).$$

Thus, the normalization factor is a marginal density at $\boldsymbol{x}_{\mathsf{s}'} = \boldsymbol{c}_{\mathsf{s}'}$. We may now compute the restriction explicitly:

$$p_{\mathsf{s}}(\boldsymbol{x}) = \frac{p(\boldsymbol{x}_{\mathsf{s}}|\boldsymbol{x}_{\mathsf{s}'})p(\boldsymbol{x}_{\mathsf{s}'})\delta_{\mathsf{s}}(\boldsymbol{x})}{p(\boldsymbol{x}_{\mathsf{s}'} = \boldsymbol{c}_{\mathsf{s}'})} = p(\boldsymbol{x}_{\mathsf{s}}|\boldsymbol{x}_{\mathsf{s}'} = \boldsymbol{c}_{\mathsf{s}'})\delta_{\mathsf{s}}(\boldsymbol{x}). \tag{4}$$

In other words, the information projection to a sparse support domain is the density of $\boldsymbol{x}_{\mathsf{s}}$ conditioned on $\boldsymbol{x}_{\mathsf{s}'} = \boldsymbol{c}_{\mathsf{s}'}$. The resulting gap is:

$$\mathrm{KL}(p_{\mathsf{s}} \| p) = \int_{\mathsf{S}} p_{\mathsf{s}}(\boldsymbol{x}) \log \frac{p_{\mathsf{s}}(x)}{p(\boldsymbol{x})} = \int_{\mathsf{S}} p_{\mathsf{s}}(\boldsymbol{x}) \log \frac{p(\boldsymbol{x})}{p(\boldsymbol{x})p(\boldsymbol{x}_{\mathsf{s}'} = \boldsymbol{c}_{\mathsf{s}'})} = -\log p(\boldsymbol{x}_{\mathsf{s}'} = \boldsymbol{c}_{\mathsf{s}'}).$$

Thus, for a given target sparsity $k$, we solve:

$$\mathsf{s}_* = \arg\max_{|\mathsf{s}|=k} J(\mathsf{s}), \quad \text{where } J(\mathsf{s}) = \log p(\boldsymbol{x}_{\mathsf{s}'} = \boldsymbol{c}_{\mathsf{s}'}). \tag{5}$$

### 3.1 Submodularity and Efficient Inference

In this section, we show that the cost function $J(\mathsf{s})$ is monotone submodular, and describe the greedy forward selection algorithm for efficient inference. Let $F : \wp(d) \mapsto \mathfrak{R}$ represent a set function. $F$ is *normalized* if $F(\emptyset) = 0$. A bounded $F$ can be normalized as $\tilde{F}(\mathsf{s}) = F(\mathsf{s}) - F(\emptyset)$ with no effect on optimization. $F$ is *monotonic*, if for all subsets $\mathsf{u} \subset \mathsf{v} \subseteq \wp(d)$ it holds that $F(\mathsf{u}) \leq F(\mathsf{v})$. $F$ is *submodular*, if for all subsets $\mathsf{u}, \mathsf{v} \subseteq \mathsf{m}$ it holds that $F(\mathsf{u} \cup \mathsf{v}) + F(\mathsf{u} \cap \mathsf{v}) \leq F(\mathsf{u}) + F(\mathsf{v})$. Submodular functions have a diminishing returns property [16] i.e. the marginal gain of adding elements decreases with the size of the set.

**Theorem 8.** *Let $J : \wp(d) \mapsto \mathfrak{R}$, $J(\mathsf{s}) = \log p(\boldsymbol{x}_{\mathsf{s}'} = \boldsymbol{c}_{\mathsf{s}'})$, and define $\tilde{J}(\mathsf{s}) = J(\mathsf{s}) - J(\emptyset)$, then $\tilde{J}(\mathsf{s})$ is normalized and monotone submodular.*

While constrained maximization of submodular functions is generally NP-hard, a simple greedy forward selection heuristic has been shown to perform almost as well as the optimal in practice, and is known to have strong theoretical guarantees.

**Theorem 9** (Nemhauser et al. [16]). *In the case of any normalized, monotonic submodular function F, the set $\mathsf{s}_*$ obtained by the greedy algorithm achieves at least a constant fraction $\left(1 - \frac{1}{e}\right)$ of the objective value obtained by the optimal solution i.e. $F(\mathsf{s}_*) = \left(1 - \frac{1}{e}\right) \max\limits_{|\mathsf{s}| \leq k} F(\mathsf{s})$.*

In addition, no polynomial time algorithm can provide a better approximation guarantee unless P = NP [17]. An additional benefit of the greedy approach is that it does not require the decision of the support size $k$ to be made at training time. As an *anytime* algorithm, training can be stopped at any $k$ based on computational constraints, while still returning meaningful results. An interesting special case occurs when the base density takes a product form.

**Corollary 10.** *Let $J(\mathsf{s})$ be defined as in Theorem 8 and suppose the base density is product form i.e. $p(\boldsymbol{x}) = \prod_{i=1}^{d} p(x_i)$, then $J(\mathsf{s})$ is linear.*

In particular, define $\boldsymbol{h} = \{p(x_i = 0) \ \forall \ i \in [d]\}$, then the solution of (5) is given by set of dimensions associated with the smallest $k$ values of $\boldsymbol{h}$.

## 4  Experiments

We present experimental results comparing the proposed sparse approximate inference projection to other sparsity inducing models. We performed experiments to test the models ability to estimate the support of the reconstructed targets and the predictive regression accuracy. The regression accuracy was measured using the coefficient of determination $\mathrm{R}^2 = 1 - \sum(\hat{y} - y)^2 / \sum(y - \bar{y})^2$ where $y$ is the target response with sample mean $\bar{y}$ and $\hat{y}$ is the predicted response. $\mathrm{R}^2$ measures the gain in predictive accuracy compared to a mean model and has a maximum value of 1. The support recovery was measured using the AUC of the recovered support with respect to the true $\mathsf{s}_*$.

The baseline models are: (i) regularized least squares (*Ridge*), (ii) least absolute shrinkage and selection (*Lasso*) [18], (iii) automatic relevance determination (*ARD*) [19], (iv) *Spike and Slab* [1, 2]. *Ridge* and *Lasso* were optimized using implementations from the *scikit-learn* python package [20]. While *Ridge* does not return sparse weights, it was included as a baseline for regression performance. We implemented *ARD* using iterative re-weighted Lasso as suggested by Wipf and Nagarajan [19]. The noise variance hyperparameter for *Ridge* and *ARD* were selected from the set $10^{\{-4,-3,...,4\}}$. *Lasso* was evaluated using the default *scikit-learn* implementation where the hyperparameter is selected from 100 logarithmically spaced values based on the maximum correlation between the features and the response. For each of these models, the hyperparameter was selected in an inner 5-fold cross validation loop. For speed and scalability, we used a publicly available implementation of *Spike and Slab* [21], which uses a mean field variational approximation. In addition to the weights, *Spike and Slab* estimates the probability that each dimension is non zero. As *Spike and Slab* does not return sparse estimates, sparsity was estimated by thresholding this posterior at 0.5 for each dimension (*SpikeSlab*0.5), we also tested the full spike and slab posterior prediction for regression performance alone (*SpikeSlabFull*).

The proposed projection approach is designed to be applicable to any probabilistic model. Thus, we applied the projection approach as additional post-processing for the two Bayesian model baselines. The first method is a projection of the standard Gaussian regression posterior (*Sparse-G*) (more details in supplement). The second is a projection of the spike and spike and slab approximate posterior (*SpikeSlabKL*). We note that since the spike and slab approximate posterior uses the mean field approximation, the posterior distribution is in product form and the projection is straightforward using Corollary 10. **Support size selection:** The selection of the hyperparameter $k$ - specifying the sparsity, can be solved by standard model selection routines such as cross-validation. We found that support size selection using sequential Bayes factors [22] was particularly effective, thus the support size was selected as the first $k$ where $\log p(\boldsymbol{y}|\mathsf{S}_{k+1}) - \log p(\boldsymbol{y}|\mathsf{S}_k) < \epsilon$.

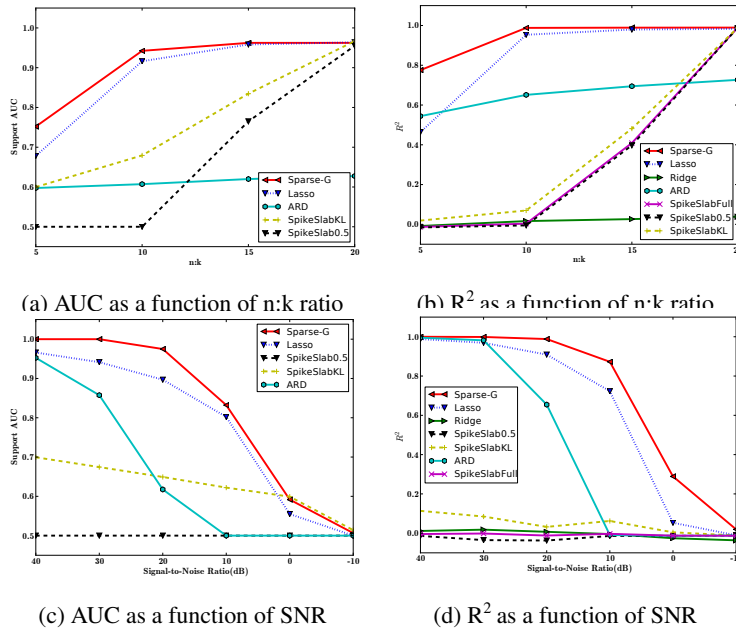

(a) AUC as a function of n:k ratio

(b) $R^2$ as a function of n:k ratio

(c) AUC as a function of SNR

(d) $R^2$ as a function of SNR

Figure 2: Simulated data performance: support recovery (AUC ) and regression ($R^2$ ).

## 4.1 Simulated Data

We generated random high dimensional feature vectors $\boldsymbol{a}_i \in \mathbb{R}^d$ with $a_{i,j} \sim \mathcal{N}(0,1)$. The response was generated as $y_i = \boldsymbol{w}^\top \boldsymbol{a}_i + \nu_i$ where $\nu_i$ represents independent additive noise with $\nu_i \sim \mathcal{N}(0, \sigma^2)$ for all $i \in [n]$. We set $\sigma^2$ implicitly via the signal to noise ration (SNR) as SNR $= \mathrm{var}(y)/\sigma^2$, where $\mathrm{var}(y)$ is the variance of $y$. In each experiment, we sampled a sparse weight vector $\boldsymbol{w}$ by sampling $k$ dimensions at random with from $[d]$, then we sampled values $w_i \sim \mathcal{N}(0,1)$ and set other dimensions to zero. We performed a series of tests to investigate the performance of the model in different scenarios. Each experiment was run 10 times with separate training and test sets. We present the average results on the test set.

Our first experiment tested the performance of all models with limited samples. Here we set $k = 20, d = 10,000$ and an SNR of 20dB. The number of training values was varied from $n = 100, \ldots, 400$ with 200 test samples. Fig. 2a shows the model performance in terms of support recovery. With limited training samples, *Sparse-G* outperformed all the baselines including *Lasso*. We also found that *SpikeSlabKL* consistently outperformed *SpikeSlab*0.5. We speculate that the significant gap between *Sparse-G* and *SpikeSlabKL* may be partly due to the mean field assumption in the underlying *Spike and Slab*. Fig. 2b shows the corresponding regression performance. Again, we found that *Sparse-G* outperformed all other baselines, with *Ridge* achieving the worst performance.

Our second experiment tested the performance of all models with high levels of noise. Here we set $k = 20, d = 10,000$ and $n = 200$ with 200 test samples. We varied the SNR from 40dB to $-10$dB (note that $\sigma^2$ increases as SNR is decreased). Fig. 2c shows the support recovery performance of the different models. We found a performance gap between *Sparse-G* and *Lasso*, more pronounced than in the small sample test. The *SpikeSlab*0.5 was the worst performing model, but the performance was improved by *SpikeSlabKL*. Only *Sparse-G* achieved perfect support recovery at low noise (high SNR ) levels. The regression performance is shown in Fig. 2d. While *ARD* and *Lasso* matched *Sparse-G* at low noise levels (high SNR ), their performance degraded much faster at higher noise levels (low SNR ).

## 4.2 Functional Neuroimaging Data

Functional magnetic resonance imaging (fMRI) is an important tool for non-invasive study of brain activity. fMRI studies involve measurements of blood oxygenation (which are sensitive to the

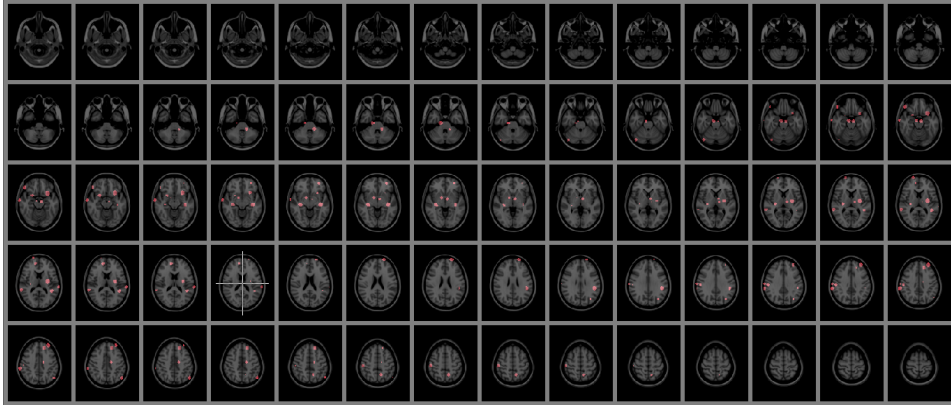

Figure 3: Support selected by *Sparse-G* applied to fMRI data with 100,000 voxels. Slices are across the vertical dimension. Selected voxels are in red.

amount of local neuronal activity) while the participant is presented with a stimulus or cognitive task. Neuroimaging signals are then analyzed to identify which brain regions which exhibit a systematic response to the stimulation, and thus to infer the functional properties of those brain regions [23]. Functional neuroimaging datasets typically consist of a relatively small number of correlated high dimensional brain images. Hence, capturing the inherent structural properties of the imaging data is critical for robust inference.

FMRI data were collected from 126 participants while the subjects performed a stop-signal task [24]. For each subject, contrast images were computed for "go" trials and successful "stop" trials using a general linear model with FMRIB Software Library (FSL), and these contrast images were used for regression against estimated stop-signal reaction times. We used the normalized Laplacian of the 3-dimensional spatial graph of the brain image voxels to define the precision matrix. This corresponds to the observation that nearby voxels tend to have similar functional activation. We present the 10-fold cross validation performance of all models tested on this data. We tested all models using the high dimensional 100,000 voxel brain image and measured average predictive $R^2$. The results are: *Sparse-G* (0.051), *Lasso* (-0.271), *Ridge* (-0.473), *ARD* (-0.478). The negative test $R^2$ for baseline models show worse predictive performance than the test mean predictor, and indicate the difficulty of this task. Even with the mean field variational inference, the *Spike and Slab* models did not scale to this dataset. Only *Sparse-G* achieved a positive $R^2$. The support selected by *Sparse-G* with all 100,000 voxels is shown in Fig. 3, sliced across the vertical dimension. The recovered voxels show biologically plausible brain locations including the orbitofrontal cortex, dorsolateral prefrontal cortex, putamen, anterior cingulate, and parietal cortex, which are correlated with the observed response. Further neuroscientific interpretation and validation will be included in an extended version of the paper.

## 5   Conclusion

We present a principled approach for enforcing structure in Bayesian models via structured prior selection based on the maximum entropy principle. The prior is defined by the information projection of the base measure to the set of distributions supported on the constraint domain. We focus on the case of sparse structure. While the optimal prior is intractable in general, we show that approximate inference using selected convex subsets is equivalent to maximizing a submodular function subject to cardinality constraints, and propose an efficient greedy forward selection procedure which is guaranteed to achieve within a $\left(1 - \frac{1}{e}\right)$ factor of the global optimum. For future work, we plan to explore applications of our approach with other structural constraints such as low rank and structured sparsity for matrix-variate sample spaces. We also plan to explore more complicated base distributions on other samples spaces.

**Acknowledgments:** fMRI data was provided by the Consortium for Neuropsychiatric Phenomics (NIH Roadmap for Medical Research grants UL1-DE019580, RL1MH083269, RL1DA024853, PL1MH083271).

## Footnotes

[1]When this condition is violated, we simply redefine $\mathsf{X}$ as the subdomain supporting p.

[2]Specifically, Williams [13] noted "Relative information has been defined only for unconditional distributions, which say nothing about the relative probabilities of events of probability zero."

[3]Where $p$ may represent the conditional densities as in Section 2.1. To simplify the discussion, we suppress the dependence on $\hat{y}$.

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
