[Supplementary Material]

# Supplement: On Prior Distributions and Approximate Inference for Structured Variables

**Oluwasanmi Koyejo**
Psychology Dept., Stanford
sanmi@stanford.edu

**Rajiv Khanna**
ECE Dept., UT Austin
rajivak@utexas.edu

**Joydeep Ghosh**
ECE Dept., UT Austin
ghosh@ece.utexas.edu

**Russell A. Poldrack**
Psychology Dept., Stanford
poldrack@stanford.edu

This supplement contains the proofs of theorems and additional exposition not included in the manuscript.

## Preliminaries

We begin additional definitions that will be useful for the proofs.

Let $\mathsf{X}$ be either a countable set, or a complete separable metric space equipped with the standard Borel $\sigma$-algebra of measurable sets, and let $\boldsymbol{Q}$ and $\boldsymbol{P}$ be probability measures on $\mathsf{X}$. The *relative entropy*, also known as the *Kullback-Leibler divergence* (KL divergence) of $\boldsymbol{Q}$ with respect to $\boldsymbol{P}$ is given by:

$$\mathrm{KL}(\boldsymbol{Q}\|\boldsymbol{P}) = \int_{\mathsf{X}} \frac{d\boldsymbol{Q}}{d\boldsymbol{P}}(x) \log \frac{d\boldsymbol{Q}}{d\boldsymbol{P}}(x) d\boldsymbol{P}(x),$$

where $\frac{d\boldsymbol{Q}}{d\boldsymbol{P}}$ is the Radon-Nikodym derivative, existence of which requires that $\boldsymbol{Q}$ is absolutely continuous with respect to $\boldsymbol{P}$. Let $\mathcal{P} \ni p$ denote the set of probability densities on $\mathsf{X}$ i.e. positive functions $p : \mathsf{X} \mapsto [0,1]$ that integrate to one.

We make the following assumption:

**Assumption 1.** *All distributions $\boldsymbol{P}$ are absolutely continuous with respect to the dominating measure $\nu$ so there exists a density $p \in \mathcal{P}$ that satisfies $d\boldsymbol{P} = p d\nu$.*

and the condition:

**Condition 2.** *The sample space $\mathsf{X}$ is a subset of Euclidean space with $\nu$ given by the Lebesgue measure. Alternatively, $\mathsf{X}$ is a countable set with $\nu$ given by the counting measure.*

Let $\mathbb{E}$ be the *expectation operator*, which we denote using densities as $\mathbb{E}_p[f] = \int_{\mathsf{X}} p(x) f(x) dx$ to simplify notation. We suppress the dependence on the random variable when the expectation and the relative entropy are clear from context. We define a *characteristic function* of the set $\mathsf{A} \subset \mathsf{X}$, denoted by $\phi_{\mathsf{A}}$ as the function $\phi_{\mathsf{A}} : \mathsf{X} \mapsto \mathfrak{R}_+$ that satisfies $\phi_{\mathsf{A}}(\mathbf{x}) > 0$ for $x \notin \mathsf{A}$, and $\phi_{\mathsf{A}}(\mathbf{x}) = 0$ otherwise e.g. $\delta_{\mathsf{X}\setminus\mathsf{A}}$ is a characteristic function of $\mathsf{A}$. Note that this definition differs slightly from the standard definition in convex analysis, where the characteristic function evaluates to $\infty$ outside the set.

## Priors for structured variables

Our first result is that the information projection a domain restricted density set is given by the restriction of the base measure to the support set. To begin, we show that the support constraint is equivalent to a particular expectation constraint.

**Lemma.** *Let $\mathcal{F}_\mathsf{A}$ be the domain restricted density set of $\mathsf{A}$, $\phi_\mathsf{A}$ be its characteristic function, and $\mathcal{G} = \{q \in \mathcal{P} \mid \mathbb{E}_q\left[\phi_\mathsf{A}(x)\right] = 0\} \subset \mathcal{P}$. Then $\mathcal{F}_\mathsf{A} = \mathcal{G}$.*

*Proof.* $[\mathcal{F}_\mathsf{A} \subset \mathcal{G}]$: Let $q \in \mathcal{F}_\mathsf{A}$, then $\mathbb{E}_q\left[\phi_\mathsf{A}\right] = 0$, thus $q \in \mathcal{G}$. $[\mathcal{G} \subset \mathcal{F}_\mathsf{A}]$: Let $q \in \mathcal{G}$, the non-negativity of $\phi_\mathsf{A}$ implies that for each $x \in \mathsf{X}$, either $q = 0$ or $\phi_\mathsf{A} = 0$. $\quad\square$

The following Lemma from Altun and Smola [1] characterizes relative entropy minimization subject to norm ball expectation constraints. For simplicity, the theorem is modified to address the special case of the result where a solution exists and the infimum is attained.

**Lemma** (Altun and Smola [1])**.**

$$\min_{q \in \mathcal{P}} \mathrm{KL}(q\|p) \text{ s.t. } \mathbb{E}_q\left[\boldsymbol{\beta}\right] = \mathbf{b}$$

$$= \max_{\boldsymbol{\lambda}} \langle \boldsymbol{\lambda}, \mathbf{b}\rangle - \log \int_\mathsf{X} p(x)e^{\langle \boldsymbol{\lambda}, \boldsymbol{\beta}(x)\rangle}dx + e^{-1}$$

*and the unique solution is given by $q_*(x) = p(x)e^{\langle \boldsymbol{\lambda}_*, \boldsymbol{\beta}(z)\rangle - G(\boldsymbol{\lambda}_*)}$ where $\boldsymbol{\lambda}_*$ is the dual solution and $G(\boldsymbol{\lambda}_*)$ ensures normalization.*

We can now show the first main result, investigating the relationship between restriction of densities and information projection subject to domain constraints.

**Theorem 3.** *Under Condition 2, the information projection of the density $p$ to the constraint set $\mathcal{F}_\mathsf{A}$ is the restriction of $p$ to the domain $\mathsf{A}$.*

*Proof.* The information projection of $p$ to $\mathcal{F}_\mathsf{A}$ is equivalent to:

$$\min_{q \in \mathcal{P}} \mathrm{KL}(q\|p) \text{ s.t. } \mathbb{E}_q\left[\phi_\mathsf{A}\right] = 0. \tag{1}$$

The solution is given by $q_*(z) = p(x)e^{\langle \lambda_*, \phi_\mathsf{A}\rangle - G(\lambda_*)}$, where:

$$\lambda_* = \arg\max_\lambda \langle \lambda, 0\rangle - \log \int_\mathbb{X} p(x)e^{\langle \lambda, \phi_\mathsf{A}(x)\rangle}dx.$$

Clearly $\lambda_* = -\infty$, thus $e^{\langle \lambda_*, \phi_\mathsf{A}(x)\rangle} \rightarrow \delta_\mathsf{A}(x)$ via standard limit arguments, so $q_* = p(x)\delta_\mathsf{A}(x)/\int_\mathsf{A} p(x)dx$. $\quad\square$

**Corollary 4.** *Consider the product space $\mathsf{X} = \mathsf{W} \times \mathsf{Y}$. Let domain constraint be given by $\mathsf{W} \times \{\hat{y}\}$ for some $\hat{y} \in \mathsf{Y}$. Under Condition 2, the information projection of $p$ to $\mathcal{F}_{\mathsf{W} \times \{\hat{y}\}}$ is given by $p(w|y = \hat{y})\delta_{\hat{y}}$.*

*Proof.* Follows directly from Theorem 3. $\quad\square$

**Theorem 5.** *Let $\pi : [n] \mapsto [n]$ be a permutation function and $\{\mathsf{C}_{\pi(i)} \mid \mathsf{C}_{\pi(i)} \subset \mathsf{X}\}$ represent a sequence of sets with non empty intersection $\mathsf{B} = \bigcap \mathsf{C}_i \neq \emptyset$. Given a base density $p$, let $q_0 = p$, and define the sequence of information projections:*

$$q_i = \arg\min_{q \in \mathcal{F}_{\mathsf{C}_{\pi(i)}}} \mathrm{KL}(q\|q_{i-1}),$$

*Under Condition 2, $q_* = q_N$ is independent of $\pi$. Further $q_* = \min_{q \in \mathcal{F}_\mathsf{B}} \mathrm{KL}(q\|p)$.*

*Proof.* Consider the case when $n = 2$, then $\pi : [1, 2] \mapsto \{[1, 2], [2, 1]\}$. From Theorem 3, we have that $q_2(x) \propto p(x)\delta_{\mathsf{C}_1}(x)\delta_{\mathsf{C}_2}(x) = p(x)\delta_\mathsf{B}(x)$ independent of $\pi$. The proof is extended to $n > 2$ by induction. $\quad\square$

We propose approximate inference via the following rule:

$$p_{\mathsf{s}_*,\hat{y}} = \arg\min_{q \in \mathcal{D} \times \mathcal{F}_{\{\hat{y}\}}} \mathrm{KL}(q(w,y)\|p_\mathsf{A}(w,y)) = \arg\min_\mathsf{S} \left[\min_{q \in \mathcal{F}_{\mathsf{S} \times \{\hat{y}\}}} \mathrm{KL}(q(w,y)\|p_\mathsf{A}(w,y))\right]. \tag{2}$$

The solution is given by $p_{\mathsf{s}_*,\hat{y}}(w,y) = p_{\mathsf{s}_*}(w|\hat{y})\delta_{\hat{y}}$.

**Corollary 6.** *Let $p_{\mathsf{s}_*,\hat{y}}(w,y)$ be the solution of (2), then the posterior distribution $p_{\mathsf{s}_*}(w|\hat{y})$ is given by:*

$$p_{\mathsf{s}_*}(w|\hat{y}) = \underset{q \in \mathcal{D}}{\arg\min}\, \mathrm{KL}(q(w)\|p_{\mathsf{A}}(w|\hat{y})) = \underset{q \in \mathcal{D}}{\arg\min}\, \mathrm{KL}(q(w)\|p(w|\hat{y})). \tag{3}$$

*Proof.* The proof follows from repeated application of Theorem 5. The first equality follows from solving (2) in two steps. The first step involves the information projection of $p_{\mathsf{A}}(w,y)$ to $\mathcal{F}_{\mathsf{A}\times\{\hat{y}\}}$. The second step is the information projection of the resulting solution $p_{\mathsf{A}}(w|\hat{y})\delta_{\hat{y}}$ to the set $\mathsf{D}\times\mathcal{F}_{\{\hat{y}\}}$. The solution is equivalent to the direct information projection of $p_{\mathsf{A}}(w,y)$ to $\mathsf{D}\times\mathcal{F}_{\{\hat{y}\}}$ as solved by (2) by Theorem 5.

The second equality follows from the projection of $p(w,y)$ to $\mathcal{F}_{\mathsf{W}\times\{\hat{y}\}}$, followed by the information projection of the resulting solution $p(w|\hat{y})\delta_{\hat{y}}$ to the set $\mathsf{D}\times\mathcal{F}_{\{\hat{y}\}}$. Since $\mathsf{D}\subset\mathsf{A}$, the equality holds by Theorem 5. $\qquad\square$

**Corollary 7.** *Let $p_{\mathsf{s}_*,\hat{y}}(w,y)$ be the solution of (2). Define the density $p_{\mathsf{s}_*}(w)$ as:*

$$p_{\mathsf{s}_*}(w) = \underset{q \in \mathcal{F}_{\mathsf{s}_*}}{\arg\min}\, \mathrm{KL}(q(w)\|p_{\mathsf{A}}(w)) = \underset{q \in \mathcal{F}_{\mathsf{s}_*}}{\arg\min}\, \mathrm{KL}(q(w)\|p(w)). \tag{4}$$

*then $p_{\mathsf{s}_*}(w)$ is the prior distribution corresponding to the Bayesian posterior $p_{\mathsf{s}_*}(w|\hat{y})$*

*Proof.* Let $p_{\mathsf{s}_*}(w)$ be the projection of $p_{\mathsf{A}}(w)$ to the set $\mathcal{F}_{\mathsf{s}_*}$, then by Corollary 4, the posterior associated with the observation of $\hat{y}$ is given by $p_{\mathsf{s}_*}(w|\hat{y})$.

The second equality follows from the projection of $p(w)$ to $\mathcal{F}_{\mathsf{s}_*}$. As $\mathsf{S}_* \subset \mathsf{A}$, the equality holds by Theorem 5. $\qquad\square$

## Priors for structured variables

The following theorem explores the submodularity of subset selection using relative entropy. The presented theorem is a special case of the result of Madiman and Tetali [2] with the application of Assumption 1. For simplicity, the theorem is modified to address the special case.

**Theorem** (Madiman and Tetali [2]). *Let $q \in \mathcal{P}$ and $p \in \mathcal{P}$ be probability densities on $\tilde{\mathsf{X}}^d$, and let $q_{\mathsf{s}}$ and $p_{\mathsf{s}}$ be their marginals on $\tilde{\mathsf{X}}^{|\mathsf{s}|}$, such that the set function $F(\mathsf{s}) : \wp(d) \mapsto [0,\infty]$, $F(\mathsf{s}) = -\mathrm{KL}(q_{\mathsf{s}}\|p_{\mathsf{s}})$ does not take the value $-\infty$ for any $\mathsf{s} \in \wp(d)$, then $F(\mathsf{s})$ is submodular.*

In the following, we show that subset selection loss function $J(\mathsf{s})$ is monotone submodular.

**Theorem 8.** *Let $J : \wp(d) \mapsto \mathfrak{R}$, $J(\mathsf{s}) = \log p(\boldsymbol{x}_{\mathsf{s}'} = \boldsymbol{c}_{\mathsf{s}'})$, and define $\tilde{J}(\mathsf{s}) = J(\mathsf{s}) - J(\emptyset)$, then $\tilde{J}(\mathsf{s})$ is normalized and monotone submodular.*

*Proof. Normalized:* By definition $0 \leq \tilde{J}(\mathsf{s}) \leq -J(\emptyset)$.

*Monotone:* Let $\mathsf{c} \subset \mathsf{s}$, then:

$$p(\boldsymbol{x}_{\mathsf{M}\backslash\mathsf{c}}) = p(\boldsymbol{x}_{\mathsf{M}\backslash\mathsf{s}}, \boldsymbol{x}_{\mathsf{M}\backslash\mathsf{s}\cap\mathsf{c}}) \leq p(\boldsymbol{x}_{\mathsf{M}\backslash\mathsf{s}})$$

*Submodular:* Consider $F(\mathsf{s}) = \log p(\boldsymbol{x}_{\mathsf{s}} = \boldsymbol{c}_{\mathsf{s}})$. $F(\mathsf{s})$ is bounded above and below.

Recall the following identity: $\mathrm{KL}(\delta_{\{a\}}\|p) = \mathbb{E}_{\delta_{\{a\}}}\left[\log\delta_{\{a\}}\right] - \mathbb{E}_{\delta_{\{a\}}}\left[\log p\right] = -\log p(x = a)$, which follows by standard limit arguments for $\mathbb{E}_{\delta_{\{a\}}}\left[\log g(x)\right] \to 0$ with an appropriate $g(x) \to \delta_{\{a\}}$. Thus, we may define $F(\mathsf{s}) = \mathrm{KL}(\delta_{\mathsf{s}}\|p)$.

Applying the theorem of Madiman and Tetali [2], it follows that $F(\mathsf{s})$ is submodular. Finally, we note that if $F(\mathsf{s})$ is submodular, so is its reflection $J(\mathsf{s}) = F([d]\backslash\mathsf{s})$. $\qquad\square$

While maximization of submodular functions is generally NP-hard, a simple greedy forward selection heuristic (Algorithm 1), has been shown to perform almost as well as the optimal in practice, and is known to have strong theoretical guarantees.

**Algorithm 1** Greedy selection, $\max J(\mathsf{s})$ s.t. $|\mathsf{s}| = k$

---

**Input:** $k, \mathsf{s} = \emptyset$
**while** $|\mathsf{s}| < k$ **do**
    **foreach** $i \in [d] \backslash \mathsf{s}$, $f_i = J(\mathsf{s} \cup i) - J(\mathsf{s})$
    $\mathsf{s} = \mathsf{s} \cup \{\arg\max f_i\}$
**end while**
**Return:** $\mathsf{s}$.

---

**Theorem 9** (Nemhauser et al. [3]). *In the case of any normalized, monotonic submodular function $F$, the set $\mathsf{s}_*$ obtained by the greedy algorithm achieves at least a constant fraction $\left(1 - \frac{1}{e}\right)$ of the objective value obtained by the optimal solution i.e. $F(\mathsf{s}_*) = \left(1 - \frac{1}{e}\right) \max_{|\mathsf{s}| \leq k} F(\mathsf{s})$.*

**Corollary 10.** *Let $J(\mathsf{s})$ be defined as in Theorem 8 and suppose the base density is product form i.e. $p(\boldsymbol{x}) = \prod_{i=1}^{d} p(x_i)$, then $J(\mathsf{s})$ is linear.*

*Proof.* Define $\mathbf{1}_\mathsf{s} \in \mathfrak{R}^d$ as the vector $(\mathbf{1}_\mathsf{s})_i = 1$ if $i \in \mathsf{s}$ and zero otherwise, and define the vector $\boldsymbol{h} \in \mathfrak{R}^d$ taking values $\boldsymbol{h}_i = p(x_i = c_i)$. When $p(\boldsymbol{x}) = \prod_{i=1}^{d} p(x_i)$, we have that $J(s) = \log p(\boldsymbol{x}_{\mathsf{s}'} = \boldsymbol{c}_{\mathsf{s}'}) = \sum_{i \in \mathsf{s}'} \log p(x_i) = \langle \mathbf{1}_{\mathsf{s}'}, \boldsymbol{h} \rangle$, a linear function. $\qquad\square$

## Gaussian linear regression with sparse structure

Consider a generative model for $n$ samples given by a linear model combined with Gaussian noise $y_i|_{\boldsymbol{w}, \boldsymbol{z}_i} = \boldsymbol{w}^\top \boldsymbol{z}_i + \epsilon$, where the response $y_i \in \mathfrak{R}$, the feature vector $\boldsymbol{z}_i \in \mathfrak{R}^d$, and the weight vector $\boldsymbol{w} \in \mathfrak{R}^d$. The weights are drawn from the zero mean Gaussian distribution $\boldsymbol{w} \sim \mathcal{N}(\mathbf{0}, \mathbf{C})$, where $\mathbf{C} \in \mathfrak{R}^{d \times d}$ is the prior covariance matrix, and its inverse $\mathbf{D} = \mathbf{C}^{-1} \in \mathfrak{R}^{d \times d}$ is the corresponding prior precision matrix. The noise is drawn from a univariate Gaussian $\epsilon \in \mathfrak{R}, \epsilon \sim \mathcal{N}(0, \sigma^2)$. We set $\lambda = \frac{1}{\sigma^2}$. Let $\boldsymbol{y} \in \mathfrak{R}^n$ represent the responses collected into a vector with $\boldsymbol{y}(i) = y_i$, and $\mathbf{Z} \in \mathfrak{R}^{n \times d}$ represent the features in matrix form, so $\mathbf{Z}(i, :) = \boldsymbol{z}_i^\top$.

As the prior and the likelihood are Gaussian, the unconstrained posterior distribution $\boldsymbol{P}(\boldsymbol{w}|\boldsymbol{y})$ is Gaussian, represented as $\mathcal{N}(\boldsymbol{\mu}, \boldsymbol{\Sigma})$, where $\boldsymbol{\Sigma} \in \mathfrak{R}^{d \times d}$ is the posterior covariance matrix, and its inverse $\boldsymbol{\Lambda} = \mathbf{S}^{-1} \in \mathfrak{R}^{d \times d}$ is the corresponding precision matrix. The posterior precision is given by $\boldsymbol{\Lambda} = \mathbf{D} + \lambda \mathbf{Z}^\top \mathbf{Z}$. The posterior mean $\boldsymbol{\mu} \in \mathfrak{R}^d$ is given by $\boldsymbol{\mu} = \lambda \boldsymbol{\Lambda} \mathbf{Z}^\top \boldsymbol{y}$. Recall that $\boldsymbol{\mu}_\mathsf{s} \in \mathfrak{R}^{|\mathsf{s}|}$ is the subvector given by $\boldsymbol{\mu}_\mathsf{s} = \{\boldsymbol{\mu}(i) \mid i \in \mathsf{s}\}$. For matrices, define $\boldsymbol{\Sigma}_{\mathsf{s},\mathsf{c}} \in \mathfrak{R}^{|\mathsf{s}| \times |\mathsf{c}|}$ as the submatrix $\{\boldsymbol{\Sigma}(i, , j) \mid i \in \mathsf{s}, j \in \mathsf{c}\}$. We also define the linear projection matrix $\mathbf{P}_\mathsf{s} \in \mathfrak{R}^{d \times |\mathsf{s}|}$, $\mathbf{P}_\mathsf{s} : \mathfrak{R}^{|\mathsf{s}|} \mapsto \mathfrak{R}^d$ by imputing zeros as missing entries.

We set the default value $c_i = 0 \; \forall \; i \in [d]$. For any fixed $\mathsf{s}$, the information projection is given by the restriction of $\boldsymbol{P}(\boldsymbol{w}|\boldsymbol{y})$ as the conditional distribution:

$$\boldsymbol{P}(\boldsymbol{w}_\mathsf{s}|\boldsymbol{w}_{\mathsf{s}'} = \mathbf{0}, \boldsymbol{y}) = \mathcal{N}(\boldsymbol{m}_{\mathsf{s}|\mathsf{s}'}, \boldsymbol{\Sigma}_{\mathsf{s}|\mathsf{s}'}), \tag{5}$$

where $\boldsymbol{m}_{\mathsf{s}|\mathsf{s}'} \in \mathfrak{R}^{|\mathsf{s}|}$, given by $\boldsymbol{m}_{\mathsf{s}|\mathsf{s}'} = \boldsymbol{\mu}_\mathsf{s} + \boldsymbol{\Lambda}_{\mathsf{s},\mathsf{s}}^{-1} \boldsymbol{\Lambda}_{\mathsf{s},\mathsf{s}'} \boldsymbol{\mu}_\mathsf{s}$. Approximate inference is applied using the convex sparse subsets, equivalent to the submodular optimization. The resulting cost function (up to constants) is given by:

$$J(\mathsf{s}) = \boldsymbol{\mu}_{\mathsf{s}'}^\top \boldsymbol{\Sigma}_{\mathsf{s}',\mathsf{s}'}^{-1} \boldsymbol{\mu}_{\mathsf{s}'} - \log|\boldsymbol{\Sigma}_{\mathsf{s}',\mathsf{s}'}| + a_1 = (\boldsymbol{\mu} - \mathbf{P}_\mathsf{s} \boldsymbol{m}_{\mathsf{s}|\mathsf{s}'})^\top \boldsymbol{\Lambda}(\boldsymbol{\mu} - \mathbf{P}_\mathsf{s} \boldsymbol{m}_{\mathsf{s}|\mathsf{s}'}) - \log|\boldsymbol{\Lambda}_{\mathsf{s},\mathsf{s}}|, \tag{6}$$

where $a_1$ is an additive constant. The second equality may be computed directly, but is most easily derived by directly solving the information projection (3) for a fixed $\mathsf{s}$. Interpreting the resulting form, we find that the subsets are selected in order to minimize the distance between the conditional mean and marginal mean vector and the determinant term adds regularity depending on the coupling between the variables.

## Additional Experiments

**Functional Neuroimaging Data**

To evaluate all the models, we applied additional dimensionality reduction to the fMRI data resulting in 10,000 voxels. The spatially constrained Ward hierarchical clustering approach of Michel et al. [4] was used for all dimensionality reduction. The predictive $R^2$ are: *Sparse-G* (0.0167), *Lasso* (-0.299), *Ridge* (-0.542), *ARD* (-0.155), *SpikeSlabFull* (-0.250), *SpikeSlab*0.5 (-0.270) and *SpikeSlabKL* (-0.183). As expected, all models (apart from *ARD*) had worse results upon dimensionality reduction. Again, the projection approach improved the performance of the *Spike and Slab* models.

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

(a) AUC as a function of n:k ratio

(b) R$^2$ as a function of n:k ratio

Figure 1: Full size simulated data performance in terms of support recovery (AUC )

(a) AUC as a function of SNR

(b) R$^2$ as a function of SNR

Figure 2: Full size simulated data performance in terms of regression performance (R$^2$ )