[Reviews · NeurIPS 2014]

Submitted by Assigned_Reviewer_7

Overview:

The paper proposes a framework for enforcing structure in Bayesian models via structured prior selection based on the maximum entropy principle. Although the optimal prior may not be tractable, the authors developed an approximation method using submodule optimization. Contructing priors with structured variables is an important topic, so this method should be able to make good impact.

Quality

The paper is technically sound. The authors present some results on both simulated data and functional MRI data to show that the methodology is effective. The strengths and limitations are also discussed.

Clarity

Overall, this paper is clearly written and well-organized.

Typeos: Page 4, line 210 and line 211, “Corollary 10” should be “Corollary 6”?
Please make sure the references have the same format.

Originality

It has enough novelty to me if all proofs are correct.

Significance

The paper presents an important topic and suggests an approach that is computationally feasible and understandable.

Summary: This is a nice work. I recommend acceptance for this paper.

Submitted by Assigned_Reviewer_8

This paper considers the problem of doing Bayesian inference when one would like to impose structural priors that restrict the support of the distribution.

It begins by motivating incorporating support constraints via the maximum entropy principle, where we would like to impose the constraint but deviate from some base distribution as little as possible, as measured by relative entropy. The distribution that optimizes this criteria is called the information projection of the base distribution onto the constraint set. The paper begins by showing that this principle leads to intuitive results in the case of support restrictions (the information projection of p onto support set A is 1{x \in A} p(x) / \int_{x \in A} p(x) ), and that the notion can be used to describe Bayesian inference, where the posterior arises when the joint distribution is projected onto the support set defined by the observations.

When we would like to perform an information projection onto the intersection of several sets, then the paper shows that this can be done by projecting onto each set in sequence in any order and the results are unchanged.

This motivates an inference procedure in the special case of sparsity restrictions, which leads to the paper's main algorithm. The idea of the algorithm is to structure inference as an inside and outside optimization, where in the inner optimization, a posterior is computed where the set of k entries allowed to be nonzero are fixed, and in the outer optimization, the set of nonzero entries is optimized over. It is shown that this optimization problem can be formulated as a submodular optimization problem, which leads the authors to propose a greedy forward selection strategy for choosing nonzero dimensions. Results against a variety of baselines show the method to work quite well.

The main pros of the paper are as follows:
- the writing is very clear, and the arguments appear to be clearly justified; things seem technically correct
- the formulation in terms of information projections is a clean and interesting way of formalizing things, providing arguments that help better understand what it means to add constraints to models
- the algorithm appears to work well

The main con is that there is a lot of technical work to get to results that are not terribly surprising. From a practical perspective, I'm not convinced that Section 2 of the paper is necessary to get to the algorithm in Section 3. It seems that we could have simply observed that the sparsity constraints can be viewed as a union of simple constraint sets, then arrived fairly naturally at the objective that is used in Section 3, and then derived the algorithm in the paper. The fact that J(s) is submodular is easy to see. So while I enjoyed reading the paper, I'm not sure I left with any new tools for imposing prior knowledge in my models.
Summary: Interesting, well-executed paper, but I was left a bit disappointed not to have gotten more in the way of methods for incorporating structural constraints into my models.

Submitted by Assigned_Reviewer_39

This paper proposes a framework for learning with prior constraints over the parameters and apply it in sparse learning with prior knowledge on the cardinality of the parameter support.

* While this paper offers a new perspective that constraint over parameters can be enforced by projecting in KL-divergence, it does not seem to lead to new interesting theoretical results or new algorithms.

* The proposed algorithm is not clearly presented. An outline, possibly in pseudocode, may be helpful.

* The experimental result shows that the proposed method (Sparse-G) outperforms all other methods but the reason is not clearly explained. Moreover, the Spike-and-slab performs significantly worse than Lasso, which is inconsistent with results in the literature. One possible reason could be the setting of the hyper-parameter.

* The writing is generally clear except:
- The concept "structure" needs to be better explained.
- Line 310 "... spike and slab does not return sparse estimates ..." is confusing as spike-and-slab is proven to be an effective
prior for sparse learning.
Summary: The paper proposes a new framework for learning with prior constraint on parameters based projecting distributions in KL-divergence. This is kind of interesting but does not seem to have any very useful theoretical or algorithmic insight.
Author Feedback
Author rebuttal: We thank the reviewers for their thoughtful responses.

Response to Reviewer_39:
* "this paper ... does not seem to lead to new interesting theoretical results or new algorithms."
We re-iterate the main contributions of the paper (with discussion of impact) - perhaps these were not emphasized enough in our submission:
1. Equivalence of (i) information projection to domain constraints and (ii) density restriction:
This result provides a principled motivation for the domain restriction approach to structured prior design, and the first explicit proof of the equivalence between Bayesian conditioning for continuous variables and information projection with data constraints.
2. Variational inference via a family of parameterized approximations: The proposed variational approximation framework differs significantly from the standard mean field approach. Development of the technique also requires novel theoretical results on repeated information projection to intersecting constraint sets. We expect that these results will be of independent interest, and anticipate applications to other constraints beyond sparsity.
3. Application to sparse support selection and submodularity: The presented algorithm may be interpreted as a novel procedure for sparsifying distributions. As expected, such a procedure is NP-hard. Fortunately, submodularity ensures scalability and effectiveness. We note that the proposed procedure is model/distribution-agnostic. In the manuscript, the procedure is applied to two very different Bayesian posteriors - Bayesian linear regression and the spike and slab, improving performance in both cases.

* "An outline, possibly in pseudocode, may be helpful."
Further information including proofs, some pseudocode, implementation details and additional experimental results were omitted due to space constraints but were included in the supplement.

* "Sparse-­G outperforms all other methods but the reason is not clearly explained."
The proposed approach focuses the posterior on regions of high probability. We conjecture that this may reduce susceptibility to spurious noise. Further, the support constraint is analogous to L_0 regularization, thus unlike alternatives such as L_1, the procedure does not introduce additional bias. We find the experimental results to be a compelling initial justification. Future work will include a theoretical analysis.

* "The concept "structure" needs to be better explained."
Structure has the standard “structured learning” connotation. Paper gives several examples.

* Regression performance: "Spike-­and­-slab performs significantly worse than Lasso, which is inconsistent with results in the literature."
You are correct in that several works (such as [1,2,25]) have shown improved performance of the spike-and-slab as compared to Lasso. This however requires careful selection/tuning of (hyper)parameters, and most pertinently have been for relatively low dimensional datasets. Specifically, available state of the art implementations of sampling-based inference did not scale to our datasets (10,000 dims and 100,000 dims), e.g. the very well crafted BayesVarSel R package does not work in this regime. In fact, we are unaware of any sampling-based results beyond 2000 dims.

As stated in the manuscript, we used the software package [21] which implements variational inference with automated hyper-parameter estimation. We conjecture that the performance gap may be a consequence of the quality of the variational approximation.

* Structure recovery: While spike­-and-­slab estimates a joint posterior probability of variable selection, our evaluation required an explicit choice of support. We emphasize that the standard estimate (averaging) will not be sparse in general. We tested support selection by thresholding the variable inclusion probabilities, typically a reasonable heuristic. Our procedure is proposed as a more effective option. We note that explicit variable selection procedures for Bayesian models remain an active area of research [26].

Response to Reviewer_7:
Thanks for your supportive and helpful comments. Suggested typos will be corrected.

Response to Reviewer_8:
* "I'm not convinced that Section 2 ... is necessary to get to the algorithm in Section 3."
Corollary 6 proves the equivalence between information projection of the unstructured and structured posterior. Without this proof, there is no guarantee that the procedure of section 3 approximates the density of interest.
The other parts of Section 2 are needed for completeness, and to connect with a broader audience (not just experts like you).

* "The fact that J(s) is submodular is easy to see.":
Our proof uses a recent result from [27] as outlined in the appendix. It is possible that a simpler direct proof exists.

* "disappointed not to have ... more in the way of methods for incorporating structural constraints into my models.
The results of Sec. 2 are quite general i.e. are not specific to a choice of distribution/model or constraints sets. We anticipate that this work will provide the foundation for a variety of novel methods. For example, we already have promising results on support selection for discrete variables - where norm based approaches are no longer applicable.

Additional references (1-24 are in the manuscript):
[25] Mohamed, Shakir, Katherine A. Heller, and Zoubin Ghahramani. "Bayesian and L1 Approaches for Sparse Unsupervised Learning." (ICML 2012).
[26] Hahn, P. Richard, and Carlos M. Carvalho. "Decoupling shrinkage and selection in Bayesian linear models: a posterior summary perspective."
[27] M. Madiman and P. Tetali. Information inequalities for joint distributions, with interpretations and applications. IEEE Trans. IT, 56(6):2699–2713, 2010.